# Extract from the Macroalgae *Ulva rigida* Induces Table Grapes Resistance to *Botrytis cinerea*

**DOI:** 10.3390/foods11050723

**Published:** 2022-02-28

**Authors:** Alon Shomron, Danielle Duanis-Assaf, Ortal Galsurker, Alexander Golberg, Noam Alkan

**Affiliations:** 1Department of Environmental Studies, Porter School of Environment and Earth Sciences, Faculty of Exact Sciences, Tel Aviv University, Tel Aviv 6997801, Israel; alon.shomron@mail.huji.ac.il; 2Department of Postharvest Science of Fresh Produce, Agricultural Research Organization (ARO), Volcani Center, Rishon LeZion 7505101, Israel; danielle.assaf@mail.huji.ac.il (D.D.-A.); ortalg@volcani.agri.gov.il (O.G.); 3Robert H. Smith Faculty of Agriculture, Food and Environment, The Hebrew University of Jerusalem, Rehovot 76100, Israel

**Keywords:** green seaweed, *Ulva rigida*, ulvan, sulfated polysaccharides, *Botrytis cinerea*, gray mold, postharvest, induced resistance

## Abstract

Fungal pathogens are a central cause of the high wastage rates of harvested fruit and vegetables. Seaweeds from the genus *Ulva* are fast-growing edible green macroalgae whose species can be found on the shore of every continent, and therefore present a resource that can be utilized on a global scale. In this study, we found that the application of ulvan extract, a sulfated polysaccharide extracted from *Ulva rigida* (1000 mg/L), elicited table grapes defense and reduced the incidence and decay area of *Botrytis cinerea* by 43% and 41%, respectively. In addition, compared to the control group at two days post-treatment, ulvan extract elicited a variety of defense-related biomarkers such as a 43% increase in the activity of reactive oxygen species, 4-fold increase in the activity of catalase, 2-fold increase in the activity of superoxide dismutase and 1.4-fold increase in the activity of chitinase. No increase was observed in phenylalanine ammonia-lyase activity, and the treatment did not affect fruit quality parameters such as the pH levels, sugar levels, and titratable acidity of grapes. These results illustrate the potential of ulvan extract to naturally induce the plant defense response and to reduce postharvest decay.

## 1. Introduction

According to the FAO, approximately one-third (over 1.3 billion tons) of the total yearly amount of food grown and produced for human consumption worldwide goes to waste. Total global losses for fruit and vegetables range between 43 and 65%. Besides wasteful practices, which may lead to unnecessary losses of edible fruit and vegetables, a major cause of wastage can be attributed to microbial decay [1]. These postharvest rots are primarily caused by pathogenic fungi. The most effective mean to control pathogenic fungi is using synthetic fungicides on pre- and postharvest fruit and vegetables [2].

Global losses to plant pathogens such as fungi account for 13% of all global crop losses, with much of the crop and financial cost owing to the fungus *Botrytis cinerea*, which is known as one of the most detrimental postharvest fungi. *B. cinerea* infects over 200 host species belonging to more than 170 agricultural plant families, which result in economic losses with estimations up to USD 100 billion/annum worldwide [1,2,3]. Heavy use of fungicides has led *B. cinerea* to become resistant to many common fungicides [2,3]. Furthermore, due to fungicide toxicity, public concern pushes to restrict fungicides for postharvest use. In response to the detrimental impact of pesticides, new intervention methods of postharvest diseases are underway, such as the utilization of pathogen-antagonistic organisms, use of natural extracts with antifungal activity, development of genetically resistant plants, induction of defense response, inhibition of ripening, and more [4,5,6]. Many of these new postharvest techniques rely on the induction of the innate defense responses found in plants [7]. Plants evolved complex chemical and structural defense mechanisms aimed to perceive, protect, and resist attacks against microorganisms. Upon stimulation, the harvested plant initiates various antifungal chemical and biological defenses [7].

The edible green seaweed of the genus *Ulva* belongs to the family of green macroalgae (*Ulvaceae*) and is one of the most common shallow-water seaweed found around the world. *Ulva* species have been shown to contain several direct antifungal compounds, such as proteins, fatty acids and aromatic compounds, many of which were suggested to have direct antifungal activities [8,9,10]. “Ulvan” is the main water-soluble, sulfur-containing polysaccharide present in *Ulva rigida* [11]. Ulvan extract has shown to be efficient against necrotrophic pathogens because it activates plants’ jasmonic acid signaling pathway [12]. In particular, ulvan extract was shown to induce a large set of defense genes upon treatment of *Medicago truncatula* leaves and to protect the plant against infection by the pathogenic fungus *Colletotrichum trifolii* [13]. Additionally, ulvan was shown to protect various plants from three different powdery mildew pathogens [14]. Ulvan has also been shown to elicit defense responses in harvested apple fruit against the phytopathogens *Penicillium expansum* and *Botrytis cinerea* [15].

Whilst ulvan has been previously shown to be an effective plant-elicitor as a foliar spray on growing plants, there are only a few studies on ulvan’s efficacy in eliciting a defense response during postharvest. Therefore, this study aimed to test the possible postharvest ability of *Ulva rigida* extracts to control gray mold. First, we screened various *Ulva rigida*-based extracts against *B. cinerea* conidia germination and hyphal growth to characterize direct hindering effects. Second, we examined the extract’s ability to elicit the harvested fruit defense response using the model of harvested table grapes interaction with *B. cinerea*. We found that the crude extractions of ethanol or acetone of *Ulva rigida* had no direct antifungal effects against the fungus *B. cinerea* in vitro. However, the acidic thermochemical hydrolysis extract (ulvan extract) did show a reduction in the gray mold incidence in vivo. The results of this study suggest that ulvan extract can elicit a defense response in postharvest grapes against the common gray mold and could potentially be used as a future approach to extend the shelf life of table grapes.

## 2. Materials and Methods

### 2.1. Preparation of Algal Extracts

#### 2.1.1. On-Shore Biomass Cultivation and Processing of *Ulva rigida*

Green leafy macroalgae *Ulva rigida* were cultivated [16], utilizing a closed macroalgae photobioreactor system with the following conditions: the growth media was natural Mediterranean seawater with a salinity range of 37–40 permille under natural irradiance. Artificial seawater at 37 permille salinity was periodically supplemented by dissolving dried Red Sea salt (Red Sea Fish Inc., Red Sea, Israel) in distilled deionized water with the addition of ammonium nitrate (NH_4_NO_3_, Haifa Chemicals Ltd., Haifa, Israel) and phosphoric acid (H_3_PO_4_, Haifa Chemicals Ltd., Haifa, Israel) to maintain 6.4 g·m^−3^ of nitrogen and 0.97 g·m^−3^ of phosphorus. CO_2_ and O_2_ were added as bubbled atmospheric air [16].

Fresh wet biomass was harvested and centrifuged in a nylon mesh bag at 4480 g (Spin Dryer CE-88, Zhenjiang, China) for 1 min to rid biomass of surface water. Harvested fresh biomass was blended with deionized water (Ninja-BL480, Needham, MA, USA) until 0.5–1 mm pieces were attained. All blended algae were pooled together in 25 L of deionized water (heated to 40 °C for 30 min). *Ulva rigida* biomass was filtered through a fine nylon mesh bag, washed, dried at 40 °C in an oven, and ground to a fine powder using a coffee grinder (47671, Morphy Richards, Yorkshire, England). The powdered biomass was divided into three groups for ethanolic, acetonic, and aqueous extractions.

#### 2.1.2. Thermochemical Extraction

Extraction procedure performed as described previously with slight modifications [11]. Powdered biomass was funneled into 1 L borosilicate bottles and filled with 90 °C deionized water until 3% *w*/*v* was attained. Next, the pH was adjusted to 2.0 using hydrochloric acid, and the bottles were placed in a water bath at 90 °C and shook in a reciprocating fashion at 2.5 Hz for 3 h. The bottles were left to cool to room temperature and filtered through fine nylon mesh bags. The filtrate was collected and centrifuged at 2970 g for 30 min (TGL-18 refrigerated centrifuge, Changsha Yingtai Instrument Co., Ltd., Changsha City, China). The collected supernatant was divided amongst 50 mL tubes and mixed with 70% ethanol cooled to −80 °C in a 1:4 (supernatant to ethanol) ratio and shaken for 15 s in a vortex. The test tubes were centrifuged at 16,050 g for 30 min at 4 °C. The supernatant salinity was measured with a conductometer, discarded, and the precipitate (“ulvan extract”) was again mixed with 70% ethanol, vortexed, and centrifuged. This process was repeated until the supernatant above the ulvan extract precipitate indicated a conductivity of <10 μS/cm. The ulvan extract precipitate was placed in a 40 °C oven until it was completely dried. Then, the dried pellet was crushed using a mortar and pestle and liquid nitrogen, and consequently freeze-dried in a lyophilizer (Freezedryer Ilshin Biobase Co., Ltd., Ede, The Netherlands).

The ulvan extraction yield was calculated in the following manner:(1)$$\mathrm{Yield}\;\%=\frac{\mathrm{Dry}\;\mathrm{weight}\;\mathrm{of}\;\mathrm{ethanol}\;\mathrm{precipitate}}{\mathrm{Dry}\;\mathrm{weight}\;\mathrm{of}\;Ulva\;rigida}\times 100$$

#### 2.1.3. Organic Solvents Extraction

The organic solvents extraction was carried out with slight modification [10]. Powdered biomass of 100 g was divided into 2 groups and funneled into 2 separate 1 L borosilicate bottles. One bottle was filled with 500 mL of absolute ethanol (EtOH) and the other bottle with 500 mL of acetone (Ethanol, Acetone, Sigma-Aldrich, Jerusalem, Israel). Each bottle was covered with aluminum foil and placed on magnetic stirring for 120 h at room temperature. 

The contents of the bottles were separately filtered through a fine nylon mesh bag, and each filtrate was placed separately in a rotary evaporator under vacuum and concentrated to 100 mL, then further concentrated at 40 °C until a dark precipitate formed (≈50 mL). The extracts from the acetone and the ethanol extractions were assumed to be in maximum solubility and were used for all experiments, henceforth referred to as “Ulva-Acetone” and “Ulva-EtOH”.

#### 2.1.4. Characterization and Chemical Composition Analysis

Fourier Transform Infrared (FT-IR) spectra of vacuum-dried ulvan extract biomass were measured in the spectral range of 400–4000 cm^−1^ (at 4 cm^−1^ resolution) using an FT-IR machine (Bruker Tensor 27 FT-IR, Bruker, Ettlingen, Germany). The identification of the thermochemical hydrolysis extract from *Ulva* sp. was compared to the extract’s features and values of previous studies, and the percentage of similarity was calculated.

Elemental analysis (CHNS) was undertaken using CHNS Analyzer (Flash2000 CHNS/O Analyzer, Thermo Fischer Scientific, Waltham, MA, USA) at Technion Chemistry Service Unit (Israel Institute of Technology, Haifa, Israel). The elemental analysis was compared to previously published data (Appendix A).

### 2.2. In Vitro Effects of Ulva rigida Extracts on B. cinerea Fungus

#### 2.2.1. In Vitro Effect of Extracts on Conidia Germination

*Botrytis cinerea* was isolated from an infected red pepper. The mycelium was routinely grown on potato dextrose agar (PDA; Difco, NJ, USA) at 23 °C. *B. cinerea* conidia were isolated from a sporulating 14-day old plate by suspension in sterile distilled water and filtering through sterile cheesecloth. The conidia were diluted with double deionized water to a concentration of 10^5^ conidia mL^−^^1^. Conidial concentration in the suspension was microscopically determined using a hemocytometer.

Germination tests with slight modification [17] were performed on microscopic glass slides. Solutions of various additives were made by mixing 0.2% Sabouraud maltose broth (SMB; 10 gr Peptone (Difco), 40 g Maltose (Caisson labs, Smithfield, UT, USA) per 1 L) with either algal extracts or water. The solutions prepared were: 0.5% *v*/*v* Ulva-EtOH extract, 0.5% *v*/*v* Ulva-Acetone extract, 0.5% *v*/*v* EtOH, 0.5% *v*/*v* acetone, 0.05% *v*/*v* ulvan extract, 0.1% *v*/*v* ulvan extract, 0.5% *v*/*v* ulvan extract, 0.2% *v*/*v* SMB. A 10 µL drop of each solution was added to a 10 µL conidial suspension of 10^5^ Conidia mL^−1^. The slides were incubated in a humid chamber at room temperature for 24 h. Each slide was photographed under a microscope (Leica DM500, ICC50 HD camera, Heerbrugg, Switzerland) 4 times at different locations at 10× magnification, and four photos of different areas of the slide at 40× magnification. The lengths of conidia germ tubes and germination percentages were measured across all photographs using ImageJ.

#### 2.2.2. In Vitro Effect on Mycelium Grown on Agar Plate

PDA Petri dishes were made with descending concentrations of extracts (for acetonic and ethanolic extracts: 0.5%, 0.25%, and 0.125% *v*/*v*). Control plates were made having the same descending concentrations of ethanol, acetone, or PDA only. All plates were made in triplicate. The PDA plate was inoculated with 1 cm^2^ piece of *B. cinerea* mycelium from a 10-day-old plate. The plates were incubated at room temperature for seven days. The mycelium diameter was measured each day.

#### 2.2.3. In Vitro Effect on Fungal Growth Kinetics

Fungal growth kinetics with slight modification [17] was measured using a 96-well plate (Bar-Naor Ltd., Petah Tikva, Israel). Algal ethanolic, acetonic, and ulvan extracts of 1% to 0.125% were diluted in 1% SMB. Then, 200 µL from each dilution was pipetted into a 96-well plate in triplicate. Conidial suspension of 10 µL (10^5^ Conidia mL^−1^) was added to each well and mixed thoroughly. “Blanks” were filled with algal dilutions without conidial suspension. The plate was incubated at room temperature, and optical density measurements at 600 nm (OD_600_) were taken every hour for 60 h using a Synergy LX plate reader (BioTek, Winooski, VT, USA). The absorbance of three wells of repeats was averaged together and was background-corrected by subtracting the average absorbance of media alone at time zero [17].

### 2.3. Measuring Fruit Response to Ulva rigida Extracts

#### 2.3.1. In Vivo Test on Grapes Inoculated with *B. cinerea*

Pathogenicity assays with slight modification were based on [18]. Similar weight, size, and with no visible surface blemishes grapes (*Vitis vinifera* cv. ‘Scarlotta’ Tali Grapes^©^, Moshav Lachish, Israel, 31°33′42″ N 34°50′34″ E) were harvested in October. In total, 150 grapes were cut at the pedicel, 1 cm above the grape flesh. The grapes were rinsed with water and dried, followed by a spray of 70% ethanol, and dried. The grapes were divided into two groups, each group was dipped for 1 min in a solution of either 1000 mg/L ulvan extract or DDW, left to dry, and kept at room temperature. Two days post-treatment (DPT), 40 grapes from each treatment group were punctured with a sterilized needle at 1 mm depth following inoculation with 10 µL conidial suspension (10^5^ Conidia mL^−1^) and incubated at room temperature for seven days in humid conditions. The decay diameter was measured every day, and the decay incidence (percentage of infected grapes) was calculated. The decayed area was calculated by assuming each grape is a perfect ellipse with an area of 235 mm^2^ (major axis: ≈20 mm, minor axis: ≈15 mm) and dividing the area of fungal growth (mm^2^) by the ellipse area and taking the total daily average for each group of treated grapes, where: *n* = number of samples.
(2)$$\%\;\mathrm{decay}\;\mathrm{area}=\;\frac{1}{n}\overset{n}{\underset{i=1}{\sum }}\;\frac{\mathrm{fungal}\;\mathrm{growth}\;}{235}\times 100$$

#### 2.3.2. Evaluation of Fruit Quality Parameters

Physiological parameters of the grapes—total soluble sugars (TSS), pH, and titratable acidity (percentage malic acid equivalence)—were tested on the first day (before treatment) and on 2 or 4 DPT, and with slight modification based on [19]. For testing of TSS, 4 un-inoculated grapes from each treatment (ulvan extract or DDW (control)) were randomly selected, placed inside sterilized gauze and collectively crushed, and the filtered juice was collected. Filtered juice of 1 mL from each tested group was placed in a Palette digital-refractometer PR-1 (Model DBX-55, Atago, Japan), and each measurement was repeated in triplicate.

For acidity determination, 7 un-inoculated grapes from every treatment (ulvan extract or DDW (control)) were randomly selected for testing in the following manner: 1 mL of pulp juice was dissolved in 40 mL double-distilled water and determined as malic acid equivalent mass using an automatic titrator (Model 719s, TitrinoMetrohm Ion Analysis Ltd., Herisau, Switzerland).

#### 2.3.3. Reactive Oxygen Species Levels

Method for determining reactive oxygen species (ROS) was based on Galsurker et al. (2020) with slight modification [19]. Five ulvan-treated grapes and 5 DDW-treated grapes were tested on 2 and 4 DPT. The grapes were submerged in 10 μM DCF (2′,7′-Dichlorofluorescein, Sigma-Aldrich, St. Louis, MO, USA) in phosphate-buffered saline (PBS) and placed on an orbital shaker at 1.6 Hz for 15 min in the dark. Followed by de-staining with PBS solution on the orbital shaker for 1 min, and the washing repeated twice.

The fluorescence was measured at 488 nm excitation and 510 nm emission using in- vivo Imaging System (IVIS^®^, PerkinElmer, Inc., Waltham, MA, USA) Results were calculated using IVIS^®^ Lumina II imaging system software (PerkinElmer, Inc., Waltham, MA, USA) and expressed in units of radiance [p/s/cm²/sr].

#### 2.3.4. Free Radical Scavenging Activity

DPPH (2,2-Diphenyl-1-picrylhydrazyl; D9132, Sigma-Aldrich, Jerusalem, Israel) was used to represent the overall antioxidant of the sample, according to the method described by Cheung et al. (2003), with slight modifications [20]. Grape peel samples were collected at 2 or 4 DPT. All samples were crushed to a powder using a mortar and pestle and liquid nitrogen. For each sample, 500 mg of powder was mixed with 3 mL of 70% methanol and shaken on an orbital shaker at 5 Hz for 3 h, after which samples were centrifuged at 2349 g for 20 min at room temperature. The supernatant was evaporated in rotorvap (RE-201D, Henan Province, China) at 25 °C for 6 h until 1 mL of the solution was left. The samples were stored at −20 °C until use.

A working stock solution of 0.012 g of DPPH was made by diluting with 50% methanol until an absorbance of 1 ± 0.02 at 517 nm was obtained. DPPH (990 mL) was mixed with 10 µL sample solutions, and the absorbance was taken at 517 nm using a spectrometer (Synergy LX Multi-Mode Reader, Biotek Instruments, Winooski, VT, USA). Control solutions were prepared as above with the addition of 10 µL water instead of sample solution. The amount of antioxidants was calculated using a calibration curve which was made by measuring the scavenging activity of ascorbic acid at various concentrations (0.01–0.5 g/L) [21]. Each treatment and concentration were tested in triplicates.

#### 2.3.5. Enzymatic Activity and Protein Assays

For all enzymatic assays, frozen grape peel samples were crushed to a powder using a mortar and pestle and liquid nitrogen. The protein content of enzyme extracts was calibrated by a dye-binding method of [22] with bovine serum albumin (BSA) as a standard. For each treatment and concentration, triplicates were made and tested before treatment and 2 and 4 DPT.

Phenylalanine ammonia-lyase (PAL) activity was determined according to Assis et al. (2001), with slight modification [23]. In total, 500 mg of frozen powder was homogenized with 2 mL of borate buffer (100 mM, pH 8.8) containing 5 mM β-mercaptoethanol and 2 mM ethylenediaminetetraacetic acid (EDTA), centrifuged for 30 min at 2349 g at 4 °C, and the supernatant collected. Then, 75 μL of supernatant was incubated with 150 μL of borate buffer (50 mM, pH 8.8) containing 20 mM L-phenylalanine for 60 min at 37 °C. After incubation time, the reaction was stopped by adding 75 μL of 1 M hydrochloric acid (HCl), and the production of cinnamate was measured at 290 nm. The specific enzyme activity was expressed as nmol cinnamic acid h^−1^mg of protein^−1^.

Chitinase activity was estimated according to [24]. A total of 500 mg of frozen powder was homogenized by 2 mL 50 mM sodium acetate buffer (pH 5.0), centrifuged for 30 min at 2349 g at 4 °C, and the supernatant collected. Then, 600 μL of supernatant was mixed with 125 μL of 2% (*w*/*v*) dye-labeled chitin azure in 50 mM sodium acetate buffer (pH 5.0) and incubated for 120 min at 40 °C. After incubation, the reaction was terminated by adding 25 μL of 1 M HCl, and the supernatant was measured at 550 nm. One unit was defined as the amount of enzyme needed to catalyze the formation of 1 nmol product h^−1^mg of protein^−1^.

Catalase (CAT) activity was estimated by the method of Beers and Sizer (1952) with a slight modification [25]. First, 500 mg of frozen powder was homogenized with 2 mL of sodium phosphate buffer (100 mM, pH 7), centrifuged for 30 min at 2349 g at 4 °C and the supernatant collected. The reaction mixture contained 50μL of supernatant, 150 μL of sodium phosphate buffer (100 mM, pH 7.0), and 50 μL of H_2_O_2_ (100 mM). The H_2_O_2_ decomposition was measured at 240 nm absorbance. The enzyme activity was expressed as units per mg protein (one unit = catalase converts 1 μmol of H_2_O_2_ per min).

Super-oxide dismutase (SOD) activity was assayed according to Giannopolitis and Ries (1977) with slight modification [26]. In total, 100 mg of frozen powder was homogenized with 200 μL sodium phosphate buffer (100 mM, pH 7.8), centrifuged for 30 min at 2349 g at 4 °C, and the supernatant was collected. A reaction solution containing sodium phosphate buffer (100 mM, pH 7.8), methionine (13 mM), nitroblue tetrazolium (NTB; 75 μM), EDTA (10 μM), and riboflavin (2 μM), was mixed in a 1:1 ratio with 100 μL of collected supernatant. The mixture was illuminated through a fluorescent lamp (60 μmol m^−2^ s^−1^) for 10 min, and the absorbance was read at 560 nm. For the blank, identical solutions were kept under the dark. The enzyme activity was expressed as unit mg^−1^ of protein. One unit was defined as the amount of enzyme that caused 50% inhibition of NBT.

The final enzymatic activity of CAT, PAL, and chitinase was calculated in the following manner:(3)$$\mathrm{CAT},\;\mathrm{PAL},\;\mathrm{Chitinase}\;\mathrm{activity}=\frac{1}{n}\overset{n}{\underset{i=1}{\sum }}\frac{A_{i,0}-A_{i,1}}{A_{i,0}}\times 100$$
and the SOD activity was calculated in the following manner: (4)$$\mathrm{SOD}\;\mathrm{activity}=\frac{1}{n}\overset{n}{\underset{i=1}{\sum }}\;\frac{\left(\frac{A_{i,0}-A_{i,1}}{A_{i,0}}\right)}{50\%}\times \frac{reaction\;volume}{sample\;volume}\times dilution\;factor$$
where: *n* = number of samples; A_i,0_ = control absorbance; A_i,1_ = sample absorbance.

### 2.4. Statistical Analysis

The data presented are averages and standard errors. A *t*-test was performed to compare two treatments. One-way or two-way analysis of variance (ANOVA) was used when applicable and post-hoc tests were conducted using Tukey’s HSD pairwise comparison and the Dunn–Šidák correction, using JMP Pro 15 (SAS Institute, Cary, NC, USA).

## 3. Results

### 3.1. In Vitro Effects of Ulva rigida Extracts on B. cinerea Growth

The effect of the *Ulva rigida* extracts on *B. cinerea* germination and growth was evaluated. Conidia of *B. cinerea* was germinated in 0.2% SMB medium with or without the addition of *Ulva rigida* extracts. The addition of the various ulvan extract concentrations had shown no effect on *B. cinerea* germination, resulting in 100% germination incidence after 24 h both in control and after the addition of the various extracts (Figure 1). Moreover, all germinated conidia in *Ulva* extracts had longer and more branched germ tubes compared to the control. The effect of the ulvan extract on the germination tube elongation was in a dose-dependent manner indicating its role as a carbon source for fungal growth (Figure 1). For all organic solutions containing algal extracts, fungal growth was always larger than their control counterparts. As the concentration of the acetone or ethanol decreases in the control plates, a more pronounced growth pattern can be seen due to the impeding effect of these organic solutions (Figure 1). At hindering concentrations of organic solutions, once algal material was present, increased fungal growth was evident (Figure 1).

Next, the effect of the crude ethanolic and acetonic extract on *B. cinerea* growth and mycelial development was examined. *B. cinerea* conidia were grown in SMB medium in the presence of acetonic or ethanolic extracts. In both the ethanolic and acetonic extracts, *B. cinerea* growth was higher than the growth in the SMB medium with the same percentage of organic solvent (Appendix A). Similar results were observed in mycelial growth on agar plate supplemented with the algal extracts (Appendix A). This suggests that the fungus probably found a nutritional value within the algal extracts leading to more pronounced germination, hyphal branching, and overall greater mycelial development (Figure 1 and Appendix A).

### 3.2. In Vivo Effects of Ulva rigida Extract on B. cinerea Growth

Preliminary work has shown that neither the ethanolic nor the acetonic extracts had any significant effects in culling disease severity and incidence when compared to the control group (Appendix A)**.** Therefore, we aimed to examine the capability of ulvan extract to induce defense response in table grapes. “Scarlotta” table grapes were disinfected and treated with 1000 mg/L ulvan extract. At 2 DPT, the grapes were punctured with a sterilized needle at 1 mm depth, followed by inoculation with 10 µL conidial suspension (10^5^ Conidia mL^−1^) and monitored at room temperature for seven days in humid conditions. The application of 1000 mg/L ulvan extract on “Scarlotta” winter grapes showed a decrease in gray-mold decay in terms of decay area and decay incidence. Grapes that were treated with ulvan extract showed a 41% decrease in decay area and a 43% reduction in incidence on the fourth day post-inoculation (DPI) compared to the control group. At 7 DPI, the group treated with ulvan extract showed an 8.8% reduction in incidence and a 48.4% reduction in mean decay area when compared to the water-treated control group (Figure 2), which may indicate that ulvan treatment induces defense response mechanisms in table grapes.

### 3.3. Induction of Fruit Defense Response

To understand which defense response pathways the ulvan treatment might affect, major defense pathways and pathogenesis-related enzymes were tested. First, ROS levels were determined. Un-inoculated grapes, which were treated with ulvan extract, showed a statistical increase in ROS levels compared to the water-treated, control group, on 2 DPT and even more on 4 DPT. Grapes treated with ulvan extract had a 43% increase in ROS compared to the DDW-treated group on day 4 (Figure 3).

Next, the effect of ulvan extract treatment was examined on the activity of several defense response-related enzymes. Treatment with ulvan extract increased the activity of grape SOD when compared to the water-treated group. While at 2 DPT the ulvan extract treated grapes displayed a non-significant increase in SOD activity compared to the control group, at 4 DPT the SOD activity was significantly higher by 2-fold in the ulvan extract treated grapes compared to the control group (Figure 4A). The overall CAT activity in the ulvan extract treated group was higher than in the control group. At 4 DPT, the CAT activity was 4-fold greater than in the control group (Figure 4B). However, the total antioxidant concentration (DPPH) was lower for the ulvan extract treated group compared to the control group (Appendix A).

Ulvan extract did not show an increase in PAL activity when compared to the control. In both treatments (ulvan extract and control), an increase was observed, peaking on 2 DPT, and decreasing by the 4 DPT (Figure 4C). Chitinase activity in grapes treated with ulvan extract revealed a 1.4-fold increase in activity 2 DPT compared to initial levels, while water-treated grapes showed almost no change in the chitinase activity 2 DPT compared to initial levels. By the 4 DPT the levels were reduced back to the initial levels with no differences between the treatments (Figure 4D). While treatment with ulvan extract induced the fruit defense response (Figure 4), it did not affect the fruit ripening (Appendix A). Grapes treated with 1000 mg/L ulvan extract did not lead to any changes in pH levels, sugar levels (TSS), nor in titratable acidity in comparison to the control, on any day post-treatment (Appendix A).

### 3.4. Characterization and Chemical Composition of Crude Extract of Ulva rigida Sulfated Polysaccharides (Infrared Spectroscopy and Elemental Analysis)

*Ulva rigida* was grown and used for several extractions (ethanol or acetone crude extraction, as well as thermochemical sulfated polysaccharides extraction). The extraction of *Ulva rigida* crude aqueous extract (ulvan extract; Figure 3, Figure 4 and Figure 5) was conducted using the thermochemical extraction method, yielding approximately 25.06% ulvan. To validate the ulvan chemical structure, infrared spectral data obtained from a sample of ulvan extract was analyzed across a range of wavelengths (400–4000 cm^−1^), using Fourier-transform infrared spectroscopy (FT-IR). The analysis provides a distinctive molecular fingerprint of the sample (Figure 5). The FT-IR analysis revealed characteristic peaks of sulfur-containing polysaccharides, and a full analysis of the spectra is available in Appendix A. For further characterization, the elemental analysis of carbon, hydrogen, nitrogen, and sulfur was performed using a CHNS Analyzer. The elemental components are presented as weight percentages out of the total dry weight (DW) of the sample (Table 1). Comparison of the elemental analysis results with similar extracts which were presented in previous studies found that the percentages of the elements of the current ulvan extract are within typical ranges for such an extract [27,28,29].

## 4. Discussion

Global postharvest losses for fruit and vegetables are estimated at 40%. Major loss is contributed to postharvest diseases, caused mainly by fungal pathogens. Although chemical fungicides are considered the most effective treatment against postharvest diseases, their long-term toxicity to the environment and human health is a major disadvantage and raises the need for new environmentally friendly solutions. Ulvan’s structure and antimicrobial functionality have been previously studied on various plant models and pathogens during the plants’ growing stage, while research on postharvest produce has been scarce. In general, most research was focused on increasing food production, while a substantially smaller focus was directed towards reducing losses [30]. Therefore, in the current study, we aimed to investigate the possible effect of different *Ulva rigida* extracts to control the pathogenic fungi *B. cinerea* growth and decay.

*Ulva rigida* was chosen due to its easy cultivation in various mediums on and offshore, with the advantage of having one of the fastest biomass growth rates amongst all photosynthesizing organisms and a rich source of biochemical functional components. *Ulva* species are one of the most common seaweeds found around the world, making *Ulva* a globally available marine resource. In addition, *Ulva* species were shown to possess a high amount and diversity of carbohydrates [31].

The first step in this study was to examine a possible direct antifungal effect of different *Ulva* extracts (ethanolic, acetonic, or aqueous extraction) on fungal conidia germination and hyphal growth. In this work, none of the *Ulva rigida* extracts demonstrated any hindering abilities on *B. cinerea* conidia germination (Figure 1 and Appendix A). Similar results of no effect on fungal germination were observed in other papers [32,33]. This is likely due to the fungus considering the algal material as a source of nutrition. Even at hindering concentrations of organic solutions, once algal material was present, an increase in fungal growth was evident (Figure 1 and Appendix A). Methanol extract from *Ulva* sp. at 1000 mg/L inhibited *Colletotrichum lindemuthianum* conidia germination while ulvan extract promoted fungal growth [10]. However, no methanolic control was conducted.

Since our data did not support the direct inhibition of fungal growth by *Ulva* extracts, we next examined whether these extracts could elicit a mechanism of fruit defense response. While neither the ethanolic nor the acetonic extracts reduced the disease severity and incidence (Appendix A), ulvan extract induced the fruit’s defense response and inhibited the disease progression. The concentration of 1000 mg/L was chosen for the ulvan extract based on previous publications as an optimum low-concentration, as higher concentrations of 5000–2000 mg/L [32,34] exhibited disease reduction and significant plant defense response similar to studies that used 1000 mg/L [13,35,36]. Whereas a lower concentration of ulvan extract at 200 mg/L was not effective and did not induce oxidative burst in wheat cells [35].

To test the ability of ulvan extract to induce a defense response in table grapes, the grapes were dipped in 1000 mg/L ulvan extract two days before infection. While industry standards for disease treatment in table grapes do not usually include liquid submersion, we found this treatment to be optimal for testing proof-of-concept effects of ulvan extracts in the lab. The grapes treated with ulvan extract decreased decay compared to water-treated grapes, indicating induction of the fruit defense response mechanisms (Figure 2). Our results are supported by previous studies, which demonstrated that ulvan extract reduces anthracnose disease incidence caused by *Colletotrichum gloeosporioides* by 40% in papaya and 30% in apples, four days post-ulvan-treatment [15,31].

It has been previously demonstrated that ulvan extract can induce both the systemic acquired resistance (SAR) and induced systemic resistance (ISR) systems and provide protection against fungal pathogens associated with either system, such as downy and powdery mildews—which are sensitive to salicylic acid-dependent defenses, and necrotrophic fungi such as *B. cinerea*—which are more sensitive to the jasmonic acid depended-defenses [37]. Moreover, after *Medicago truncatula* was treated with ulvan extract, an array of transcripts involved with defense response were upregulated [13]. One of the first lines of defense in the plant immunity response is the production of ROS. However, while the link between ROS generation and induced resistance to phytopathogens has been demonstrated [15], high levels of ROS can lead to faster fruit ripening and senescence, and even decay [38]. In the current study, ulvan-treated grapes display higher ROS levels at 2 and 4 DPT than water-treated grapes, indicating activation of defense-related pathways (Figure 3). To keep their cells from being oxidatively damaged, plants evolved ROS antioxidants enzymes such as SOD and CAT. Indeed, in our experimental setup, ulvan extract treatment increased SOD and CAT enzymatic activity (Figure 4). The increase in SOD and CAT enzymatic activity is consistent with [37]. While ulvan-treated grapes exhibited lower levels of total antioxidant at 2 and 4 DPT (Appendix A), some antioxidants (SOD, CAT) levels were higher than the control group. The hypothesis that ulvan may be able to downregulate certain enzymatic activities while upregulating others requires further studies to be undertaken. 

As a result of defense response induction, a global defense response is activated including hundreds of genes. Chitinase and PAL are an example of two key enzymes in plant “defense response” to pathogenic stress [39,40,41]. Ulvan extract treatment increased chitinase activity 2 DPT (Figure 4). The increase in chitinase activity in response to the ulvan treatment could be explained by ulvan’s structural similarity to phytopathogenic chitin monomers, which are secreted during the host–pathogen interaction leading to activation of pathogen-associated molecular patterns (PAMP) and ROS generation. However, the ulvan extract did not induce PAL activity. Polysaccharide extract depolymerized to oligo-ulvan had a higher PAL activity in tomato seedling leaves, but no activity was observed when the extract was de-sulfated [15]. The elemental analysis of our extract (Table 1) indicated low levels of sulfur compared to previously published data, which might explain the lack of increase in PAL activity. *B. cinerea* is known to tolerate high levels of ROS, however, it was inhibited after treatment with ulvan on fruit. Therefore, other fruit defense-related activities such as PAL, chitinase, and others are the probable cause of this fungal inhibition. In all, this study demonstrates that various extractions from *Ulva rigida* have no direct antifungal activity and only the thermochemical extraction of sulfated polysaccharides had an effect on the fruit defense response, which led to inhibition in decay development. Furthermore, this work lay down the basis for the ulvan extract mode of action to induce defense response.

To better characterize the ulvan extract, FT-IR spectra and the elemental analysis were performed and compared to published data. Those comparisons resulted in high similarity to other ulvan extraction and were shown to be within typical ranges of similar extractions (Figure 5 and Table 1). A high nitrogen content could correspond to both high protein content and non-protein nitrogenous substances, such as pigments, nucleic acids, and inorganic nitrogen. Importantly, the elemental analysis revealed that our nitrogen content was the lowest compared to other papers that analyzed the elemental constituents in ulvan extract (Table 1). We, therefore, infer that our aqueous extraction as ulvan extract is composed mostly of polysaccharides. However future studies that aim to test ulvan’s defense-inducing capabilities should provide a more in-depth characterization of ulvan, such as its monosaccharidic profile, molecular weight, and level of sulfation in order to better characterize the most active component.

Since most biocontrol methods are well below 100% efficacy, a single biocontrol approach will not meet commercial demands. Therefore, an integration of multiple methods may prove valuable and effective. Some compounded methods include combining fungicides with commercial sanitizers or salts, essential oils, altered atmosphere, plant elicitors, antagonistic organisms, and even physical methods such as temperature changes, radiation, pressure, and sonication. Many of these composite methods could have additive and even synergistic effects [42].

## 5. Conclusions

*Ulva* is a very common seaweed that was shown to have a wide range of agricultural and biomedical activities. This research tested the effect of several green seaweed *Ulva rigida* extracts on their efficacy in protecting harvested table grapes from the fungal pathogen *B. cinerea.* Our results show that ethanol and acetone-based extracts of *Ulva rigida* did not directly inhibit the conidia germination and fungal growth of *B. cinerea*, nor did they elicit an indirect inhibition effect by activating innate defense mechanisms in ‘Scarlotta’ table grapes.

While most of the previous studies focused on protecting growing crops, this study demonstrated that sulfated polysaccharides extracted from *Ulva rigida* (ulvan extract) reduced decay in harvested grapes by activating the fruit’s defense response. The study of ulvan’s mode of action in harvested fruit showed induction in various defense-related biomarkers as an increase in ROS and enzymatic activities of catalase, superoxide dismutase, and chitinase. The treatment did not affect fruit quality parameters such as the pH, sugar levels, and the titratable acidity of the grapes. Therefore, *Ulva*
*rigida* extract of sulfated polysaccharides has the potential to be used as a postharvest treatment to reduce decay development. Ulvan may be combined with other treatments in the future to achieve better decay control.

## Figures and Tables

**Figure 1 foods-11-00723-f001:**
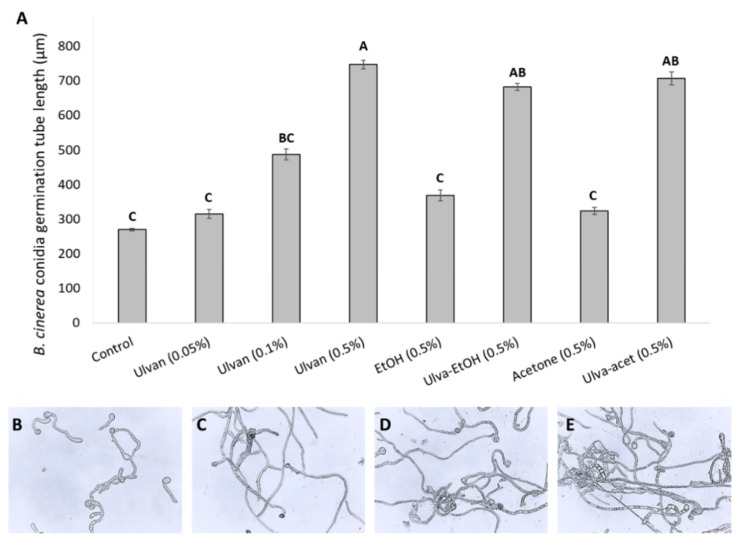
*Ulva* extracts promote in vitro growth of *B. cinerea* in Sabouraud maltose broth medium. Conidia of *B. cinerea* were germinated on glass in the presence of SMB medium and ulvan or different acetonic (acet) or ethanolic (EtOH) algal extracts in various concentrations in humid conditions at 23 °C for 24 h. (**A**) Length of *B. cinerea* conidia germination tube. Average and SE are presented. Different letters indicate a statistical difference (*p* ≤ 0.05). (**B**–**E**) Representative pictures of *B. cinerea* conidia germination after 24 h. (**B**) Control (0.2% SMB), (**C**) 500 mg/L ulvan extract, (**D**) 500 mg/L Ulva-EtOH, (**E**) 500 mg/L Ulva-Acetone.

**Figure 2 foods-11-00723-f002:**
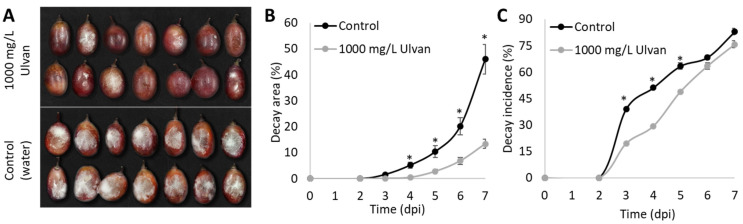
*Ulva rigida* crude aqueous extract (ulvan extract) decreases gray mold on grapes. “Scarlotta” grapes were treated with either 1000 mg/L ulvan or double deionized water (control), followed by inoculation with *B. cinerea,* two days post-treatment (DPT). (**A**) Representative picture of gray-mold decay caused by *B. cinerea* inoculation on grapes treated with 1000 mg/L ulvan extract or water (control), four days post-inoculation (DPI). (**B**) Decay area (% of fruit side). (**C**) Decay incidence (% of inoculated fruit). Asterisks indicate statistical difference between treatments at each time point (*p ≤* 0.05).

**Figure 3 foods-11-00723-f003:**
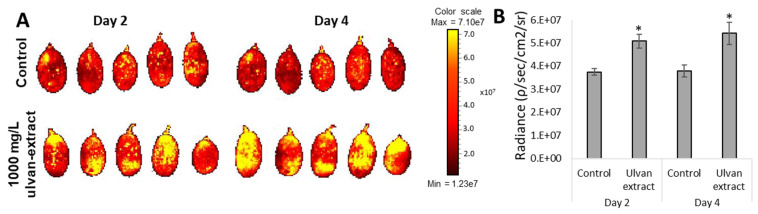
Changes in reactive oxygen species (ROS) levels. “Scarlotta” grapes were treated with 1000 mg/L ulvan extract or double deionized water (Control). The fruit was dyed with DCF, and the fluorescence (radiance [p/s/cm²/sr]) was detected using an In vivo Imaging System (IVIS^®^). (**A**) Average radiance was quantified on 2 and 4 DPT. Asterisks indicate statistical significance between treatments *(**p* ≤ 0.05). (**B**) Fluorescence was visualized on 2 and 4 DPT.

**Figure 4 foods-11-00723-f004:**
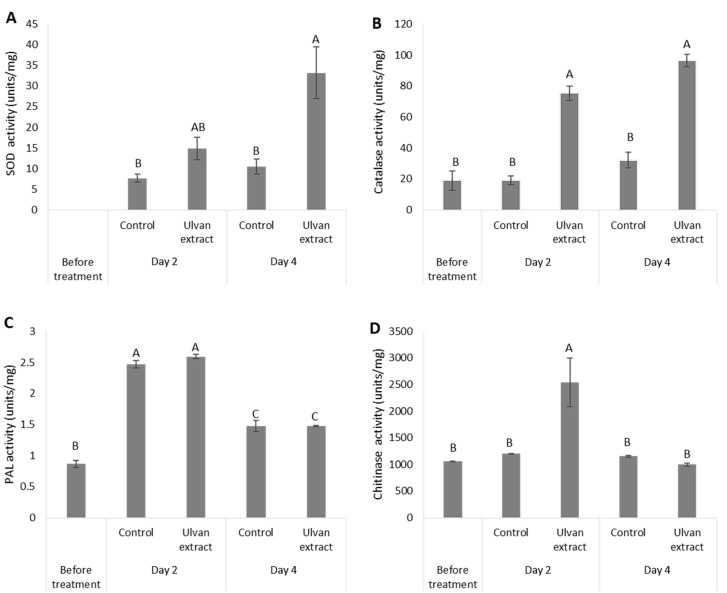
Changes in fruit enzymatic activity of plant-defense biomarkers. Enzymatic activity of (**A**) superoxide dismutase (SOD), (**B**) catalase (CAT), (**C**) phenylalanine ammonia-lyase (PAL), and (**D**) chitinase in uninoculated “Scarlotta” grapes treated with 1000 mg/L ulvan extract or water (control) on two- and four-days post-treatment (DPT). Different letters indicate as difference (*p* ≤ 0.05).

**Figure 5 foods-11-00723-f005:**
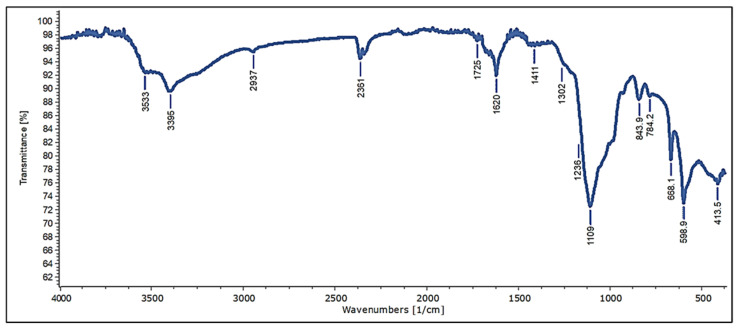
The Fourier-transform infrared spectrum (FT-IR). Protruding peaks correspond to chemical structures found within the sample of ulvan extract, and the numbers correspond to the vibration frequency of the chemical structures. Larger peaks show a greater abundance of these structures within the sample.

**Table 1 foods-11-00723-t001:** Thermochemical acidic hydrolysis extract of *Ulva rigida* (ulvan) elemental composition. Analysis of carbon (C), hydrogen (H), nitrogen (N), and sulfate (S) in the current study compared to the literature.

% of DW	C	H	N	S
Ulvan extract in the current study ^#^	22.74	4.61	0.63	7.00
Costa et al., 2012 [27]	22.80	4.30	1.00	8.40
del Rocío Quezada-Rodríguez and Fajer-Ávila, 2017 [28]	19.47	3.78	0.89	8.69
Lahaye and Robic, 2007 [29]	19.35	3.88	0.98	5.62

^#^ Data display the average of three repeats.

## Data Availability

The datasets generated for this study are available on request to the corresponding author.

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
