# Peer review of "Extract from the Macroalgae *Ulva rigida* Induces Table Grapes Resistance to *Botrytis cinerea"

_foods, 2022, doi:10.3390/foods11050723_

Round 1
Reviewer 1 Report
This is an interesting research of great importance to the agricultural industry especially for postharvest processing.
Authors need to improve the title to represent the entire work. Proposed title provided.
Pay attention to the context some terminologies are used. Italics are not used in parts required. Superscript and subscript not used correctly in some parts. In some cases, statements are out of context or contradicts to the actual meaning. Please refer to comments on manuscript.
In most parts of the methodology, references are not cited.
Section 2.1.3 Please pay attention to the type of extract. This section is poorly written and is confusing. See comments
Authors tend to use non-scientific context in some parts. Please thoroughly go through the manuscript for proper english and ensure the content is written direct to the point.
I am not familiar with the enzymatic experiments.
Eventually, I suggest the authors to highlight how does this work stand out compared to pre-existing postharvest treatment and how will this such findings be able to improve them.

Author Response
Thank you for reviewing our manuscript
Comment: Authors need to improve the title to represent the entire work. Proposed title provided.
Reply: The title was changed accordingly.
Comment: Pay attention to the context some terminologies are used. Italics are not used in parts required. Superscript and subscript not used correctly in some parts. In some cases, statements are out of context or contradicts to the actual meaning. Please refer to comments on manuscript.
Reply: All Italics, Subscripts, and Superscripts were corrected.
Comment: In most parts of the methodology, references are not cited.
Reply: Additional references provided for methodologies.
Comment: Section 2.1.3 Please pay attention to the type of extract. This section is poorly written and is confusing. See comments.
Reply: Sections 2.1.3 have been re-written to be clearer.
Comment: Authors tend to use non-scientific context in some parts. Please thoroughly go through the manuscript for proper English and ensure the content is written direct to the point.
Reply: Dangling's writing was corrected. Informal language was formalized.
Comment: Eventually, I suggest the authors to highlight how does this work stand out compared to pre-existing postharvest treatment and how will this such findings be able to improve them.
Reply: The innovation of this study was elucidated in the discussion (L528-32) and the conclusion was rewritten.
Reviewer 2 Report
In summary, the authors tested the application of aqueous seaweed extract (1,000 mg/L) as a biocontrol against B. cinerea. As the extract had no antifungal action, on the contrary, it increased the germination and growth capacity of the fungus, the authors resolved to study the alterations caused by the treatment on enzymes related to the defence mechanism of table grapes.
The introduction provides sufficient background information and the objectives are clear and well ordered. Materials and methods section and statistical parameters are well explained but researchers could update the references in this section. However, I have some questions and suggestions that I would like to address.
Major comments:
Materials and methods
-sections 2.1.2 and 2.1.3: why did the authors use different extraction processes for the water and the alcohol/acetone extract? Why did you use different temperatures and biomass concentrations? It is difficult to compare the extracts if they are obtained by such different methodologies. Have you considered the possibility of making an aqueous extract using the same extraction process as the ethanol and acetone extracts? Why were not all extracts freeze-dried? How can I be sure that no ethanol remains after freeze-drying the aqueous extract? Have you performed a control group?
-Explain the acronyms DPT and DPI for the first time they appear in the text.
Minor comments:
-According to SI Units add a space between the number and the symbols ºC and %.
-Line 303: delete double space
-Line 320: 23 ºC
- Line 426-430: reference
-Line 450- 453: If other authors have already tested Ulva extracts at 1000 mg/L, why haven't you tested higher concentrations? If possible, include the answer in the discussion.
-Include in discussion: What is the reason why the control has more antioxidants than the treated group? Could the authors discuss this question? Also in Figure s3 I am not sure if there is a significant difference between the groups if the values presented are means +/- SE. Can the authors check the statistics?
Author Response
Thank you for reviewing our manuscript. All your suggestions were integrated into the manuscript.
Comment: sections 2.1.2 and 2.1.3: why did the authors use different extraction processes for the water and the alcohol/acetone extract? Why did you use different temperatures and biomass concentrations? It is difficult to compare the extracts if they are obtained by such different methodologies. Have you considered the possibility of making an aqueous extract using the same extraction process as the ethanol and acetone extracts? Why were not all extracts freeze-dried? How can I be sure that no ethanol remains after freeze-drying the aqueous extract? Have you performed a control group?
Reply: Sections 2.1.2 and 2.1.3 have been re-written to be clearer.
The different extraction methods were used depending on the purpose of the extract. By using an organic solvent, we aimed to extract all compounds that can be dissolved in ethanol or acetone while the water extraction aimed to extract mainly ulvan. The extraction methods are based on previous studies and were designed for the best yield of extraction. We agree with the reviewer that the different extraction methods make it harder to compare the different extractions, However, since the methods are similar to previous studies using Ulva extract we were able to compare them to previous studies.
To be sure that no ethanol remains, the aqueous extract was precipitated in a centrifuge, then heated in an oven to evaporate the ethanol and reach a complete drying, later the pellet was further dried with a lyophilizer. To test the purity of the extract we used FT-IR, which showed no ethanol remains.
Comment: Explain the acronyms DPT and DPI for the first time they appear in the text.
Reply: Acronyms are fixed and explained as they first appear in the paper.
Comment: According to SI Units add a space between the number and the symbols ºC and %.
Reply: Spaces were added between the symbols ºC and %.
Comment: If other authors have already tested Ulva extracts at 1000 mg/L, why haven't you tested higher concentrations? If possible, include the answer in the discussion.
Reply: In "Discussion", the rationale for choosing 1000 mg/L was further elucidated and additional references were added. Lines 475-484.
Comment: What is the reason why the control has more antioxidants than the treated group? Could the authors discuss this question?
Reply: In the Discussion, the reason for higher levels of antioxidants in the treated group was articulated. Lines 510-513.
Comment: in Figure s3 I am not sure if there is a significant difference between the groups if the values presented are means +/- SE. Can the authors check the statistics?
Reply: Statistical differences between the presented group mean in Figure S3 were tested using the post-hoc T-test and have shown to be statistically significant from one another.
Additional amendments:
- Terminologies were corrected and aligned in ‘Abstract’.
- In section 2.1.4., spectrophotometer was corrected to FT-IR machine.
- In section 2.2.4., methods for choosing optical density parameter was referenced.
- In section 2.3.2., methods for the evaluation of fruit quality parameters have been clarified.
- Conclusions clarified.
- References standardized.